# Large Language Models are biased to overestimate profoundness

**Eugenio Herrera-Berg**[*1]**, Tomás Vergara Browne**[*1,2]**, Pablo León-Villagrá**[†3]**,**
**Marc-Lluís Vives**[†4]**, Cristian Buc Calderon**[†1]

[1]Centro Nacional de Inteligencia Artificial (CENIA), Chile
[2]Pontificia Universidad Católica de Chile, Chile
[3]Brown University, United States
[4]Leiden University, the Netherlands

## Abstract

Recent advancements in natural language processing by large language models (LLMs), such as GPT-4, have been suggested to approach Artificial General Intelligence. And yet, it is still under dispute whether LLMs possess similar reasoning abilities to humans. This study evaluates GPT-4 and various other LLMs in judging the profoundness of mundane, motivational, and pseudo-profound statements. We found a significant statement-to-statement correlation between the LLMs and humans, irrespective of the type of statements and the prompting technique used. However, LLMs systematically overestimate the profoundness of nonsensical statements, with the exception of Tk-instruct, which uniquely underestimates the profoundness of statements. Only few-shot learning prompts, as opposed to chain-of-thought prompting, draw LLMs ratings closer to humans. Furthermore, this work provides insights into the potential biases induced by Reinforcement Learning from Human Feedback (RLHF), inducing an increase in the bias to overestimate the profoundness of statements.

## 1   Introduction

GPT-4 has now achieved the ability to perform a wide range of tasks on par with humans (OpenAI, 2023). Given that current LLMs excel at text interpretation and generation, recent work has assessed LLMs' ability to perform tasks beyond simple language understanding. And the evidence for LLMs' ability to perform these tasks is changing rapidly: while previous versions of LLMs (e.g., GPT-3.5) fell short in tasks seen as unsolvable with textual input alone (Ullman, 2023), more recent models, such as GPT-4, have been suggested to achieve them (Bubeck et al., 2023). However, most of these studies have focused on presenting the LLM with statements following conversational maxims.

Thus, when presenting prompts to an LLM, one generally strives to make the statement informative and truthful to convey one's ideas or goals successfully. However, not all language use is aimed at efficiently communicating information. Sometimes, it can be beneficial for a speaker to obscure the meaning of an utterance, for example, to deceive or persuade or to hide one's true intentions. Successful, human-level communication requires the listener to detect such language use; otherwise, they are susceptible to deception. Here, we assess whether GPT-4 and other LLMs can identify language created with the aim of impressing the listener rather than communicating meaning.

The term "pseudo-profound bullshit" (BS) refers to sentences that seem to have a deep meaning at first glance, but are meaningless (Pennycook et al., 2015). These sentences are syntactically correct and presented as true and significant but, upon further consideration, lack substance. For example:

> "Consciousness is the growth of coherence, and of us."

Pseudo-profound bullshit is thus an example of language use that is not aimed at conveying information but evoking an interpretation in the listener in order to seem meaningful and insightful; and could potentially fool LLMs to produce non-desired responses. Cognitive strategies deployed during language exchanges to uncover (and create) these strategies require sophisticated skills (Musker and Pavlick, 2023), such as making recursive inferences or assessing the actual meaning of hard-to-parse sentences (Bubeck et al., 2023). In fact, in humans, the ability to detect the shallowness of pseudo-profound bullshit correlates with classical measures of cognitive sophistication like verbal intelligence or individual tendencies to exert deliberation (Pennycook et al., 2015).

Because LLMs perform better in tasks that re-

---

[*]Equal contribution.
[†]Co-senior authors.

quire knowledge of syntax, morphology, or phonology while struggling in tasks that require formal reasoning (e.g., logic), causal world knowledge, situation modeling, or communicative intent (Mahowald et al., 2023), it is an open question whether an LLM will be able to detect the presence of pseudo-profound bullshit.

We find that GPT-4, and most other LLMs tested here, display a strong bias towards profoundness, i.e., the pseudo-profound statements are systematically ranked above a mid-point level of profoundness. In contrast, humans rank these statements below this mid-point level. The one exception we found is with Tk-Instruct, which consistently underestimates the profoundness of every statement, including that of motivational statements.

Moreover, we show that chain-of-thought prompting methods (Wei et al., 2022), which typically increase the reasoning abilities in LLMs, have no statistical effect on this ranking; and that only few-shot learning allows GPT-4 to rate statements more similar to humans and below the mid-point ranking. Finally, we show that, despite the biases found, the statement-to-statement rankings display a strong correlation between the LLMs and humans.

## 2   Related Work

**Probing Language Understanding.** LLMs can be treated as participants in psycholinguistic (Linzen et al., 2016; Dillion et al., 2023) or cognitive science experimental studies (Binz and Schulz, 2023). Thus, recent work suggests a series of diagnostics to analyze LLMs inspired by human experiments (Ettinger, 2020). For instance, LLMs can represent hierarchical syntactic structure (Lin et al., 2019) but tend to struggle in semantic tasks (Tenney et al., 2019) or display similar content effects as humans (Dasgupta et al., 2022). Furthermore, Hu et al. (2023) compared a variety of LLMs to human evaluation on seven pragmatic phenomena to test possible correlations between human and models' judgments.

**Boosting Reasoning in LLMs.** Recent work evaluates the reasoning abilities in LLMs by analyzing the techniques that elicit reasoning (Huang and Chang, 2022). For instance, several prompting methods increase the reasoning abilities of LLMs. Chain-of-thought (CoT) prompting is a method that implements a sequence of interposed natural language processing steps leading to a final answer

(Wei et al., 2022; Zhao et al., 2023; Lyu et al., 2023). Furthermore, increasing reasoning abilities can also be elicited via Zero-shot CoT by simply adding the sentence "let's think step by step" at the end of the prompt (Kojima et al., 2022). In the same vein, showing a few examples to LLMs allows them to quickly learn to reason on complex tasks (Brown et al., 2020; Tsimpoukelli et al., 2021), an ability known as few-shot learning.

## 3   Method

Our study is based on previous research assessing whether humans are receptive to pseudo-profound statements (Pennycook et al., 2015). In the original study by Pennycook et al., 198 humans ranked the profoundness of statements on a 5-point Likert scale: 1 = Not at all profound, 2 = somewhat profound, 3 = fairly profound, 4 = definitely profound, 5 = very profound.

We replicate this study using a variety of LLMs to judge the profoundness of statements. The LLMs used are GPT-4 and various other models (Flan-T5 XL (Chung et al., 2022), Llama-2 13B with and without RLHF (Touvron et al., 2023), Vicuna 13B (Chiang et al., 2023) and Tk-Instruct 11B (Wang et al., 2022)). We performed 20 repetitions for each experiment to account for the non-deterministic nature of token generation (given that we use non-zero temperature). We investigate how sensitive this model is to pseudo-profound bullshit by systematically comparing it with human ratings on these same and similar statements. Furthermore, we use distinct prompting methods that have been shown to increase the reasoning abilities of LLMs (see below). All of our code and results are publicly available[1].

### 3.1   Dataset

We generated five distinct datasets of sentences. For dataset 1, we used the same 30 pseudo-profound BS statements as those used in experiment 2 of Pennycook et al. (2015). For dataset 2, we built a dataset comprising 30 novel pseudo-profound BS statements generated following the same procedure as Pennycook et al. (2015). Dataset 3 was generated on the basis of the first dataset, but we generated 30 novel statements by switching words from one sentence to another but maintaining the syntactic structure (dataset 3; see Appendix

---

[1]https://github.com/ouhenio/llms-overstimate-profoundness

A for further details). As points of comparison, we also used ten mundane (non-profound, dataset 4) and ten motivational (profound, dataset 5) statements from Pennycook et al. (2015) paper. An example from each dataset can be seen in Appendix A (Table 1).

## 3.2 Experimental Design

We tested the LLMs on the five datasets described in the previous section. The LLMs replied on the same Likert scale used in Pennycook et al. (2015). We further manipulated two factors: the prompt method, and the temperature. We tested three distinct prompting methods: the original instruction participants received in Pennycook et al. (2015) (prompt 1), few-shot learning (Brown et al., 2020) (prompt 2-3), and chain-of-though (CoT) prompting (Wei et al., 2022) (prompt 4-10). All the prompts are listed in Appendix A (Table 2). The temperature was set to 0.1 and 0.7 to assess the effect of variability in the outputs.

We let the LLMs predict one token corresponding to its rating of the profoundness of the sentence, and parse it as an integer.

## 3.3 Statistical Analyses

We performed an item analysis by averaging ratings across human subjects and the LLMs' responses for each statement. Given that our goal was to compare each LLM to human-level performance, we ran a 3 (between-items) × 2 (within-items) two-way mixed ANOVA, for each LLM (on the original Pennycook et al. (2015) prompt). The between-subject factor was statement type (mundane, motivational, or pseudo-profound BS), and the within-subject factor was agent type (human or LLM). To assess a potential profoundness bias when rating pseudo-profound bullshit, we tested the ratings against the mid-point scale. To analyze a potential prompt effect, we first focused on GPT-4's BS statement ratings, and performed a two-way ANOVA (with prompts and statement types as factors). Moreover, to compare the effect of prompting between models, we performed a two-way ANOVA (with model type and prompts as factors) on the LLMs' BS statements ratings.

Finally, we ran regressions to test the degree to which human ratings were predicted by the LLMs. Human mean ratings for each statement was predicted by each LLM's ratings, with statement type (mundane, motivational, or pseudo-profound bullshit), evaluation prompt (1 to 10), and temperature

as main effects.

Since manipulating the temperature of the LLMs did not significantly change the resulting ratings, we collapsed ratings across temperatures in all reported analyses. Moreover, given that we found no differences in ratings between datasets 1, 2, and 3, we did not include the ratings of datasets 2 and 3 in the analyses. Lastly, for simplicity and brevity, we only report statistical results of GPT-4; results of other LLMs can be found in Appendix A (Table 3)

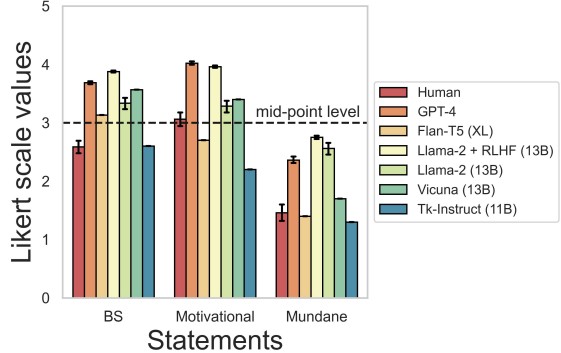

Figure 1: Distribution of ratings per statement type in humans and the LLMs.

## 4 Results

In the original study by Pennycook et al., humans discriminated between mundane, pseudo-profound BS, and motivational statements. Unsurprisingly, mundane statements were rated as the least profound, motivational statements as the most profound, and pseudo-profound BS was rated in between (Figure 1). This trend was also captured by the LLMs' ratings (Figure 1, the main effect of statement type: $F(2,47) = 72.09$, $p < 0.001$). Importantly, even though the LLMs followed the same qualitative pattern as humans, most of LLMs' ratings of profoundness were constantly higher than human ratings (see Figure 1, main effect of agent type: $F(1,47) = 185.28$, $p < 0.001$). The bias towards profoundness observed was independent of whether the statement was mundane, pseudo-profound BS, or motivational (non-significant interaction: $F(2,47) = 0.96$, $p = 0.40$) for GPT-. For the rest of LLMs, profoundness was overestimated only in pseudoprofound BS statements (Flan-T5 (XL), Vicuna (13B)) or it was larger for those statements (Llama-2 + RLHF (13B), Llama-2 (13-B)), as denoted by a significant interaction between statement and agent types (all $ps < 0.001$, see Table

3 in the Supplement). Tk-Instruct showed the opposite effect, but at the cost of underestimating the profoundness of motivational statements. We included a possible explanation for these phenomena in the discussion.

We used 10 prompts to obtain each LLM's ratings. Results reveal that prompting significantly influenced the ratings for GPT-4 (main effect of the prompt: $F(9,423) = 18.12$, $p < 0.001$). Crucially, the effect of prompting showed a significant interaction with statement type (interaction effect: $F(18, 423) = 35.06$, $p < 0.001$). Further analyses revealed that the interaction was driven by n-shot learning evaluation prompts, which lowered the ratings for pseudo-profound BS statements. Planned t-test between the two n-shot learning and other prompts revealed a significant decrease in the rating for the pseudo-profound BS (see Figure 2, all $ps < 0.001$, Bonferroni-corrected), while n-shot learning prompts did not affect the mundane or motivational statements (all $ps > 0.05$). Importantly, the ratings from the n-shot learning prompts were not significantly different than the human ratings for pseudo-profound BS (all $ps > 0.05$). Other LLMs were also affected by prompting, but significantly less (interaction effect of the two-factor ANOVA contrasting LLMs with evaluation prompt: $F(45, 1305) = 21.52$, $p < 0.001$, see Figure 2).

In addition to assessing profoundness ratings for pseudo-profound statements relative to other statement types, we investigated the absolute rating on the scale. Due to the structure of the response scale, ratings below three are considered not to be profound. Thus, if LLMs detect pseudo-profound BS, their ratings should be significantly lower than the mid-point level of 3. Note that in Figure 3, we mean-center the rating data to facilitate data interpretation; above 0 implies overestimation, and below 0 implies underestimation with respect to the mid-point level.

Our results reveal that, overall (with the exception of TK-instruct), LLMs' are biased towards overestimating the profoundness of pseudo-profound bullshit (ratings were significantly higher than 3: $t(29) = 6.67$, $p < 0.001$, average = 3.70, see Figure 3). This is in contrast with humans, who perceived pseudo-profound bullshit statements as non-profound (ratings were significantly lower than 3: $t(29) = -9.71$, $p < 0.001$, average = 2.56, Figure 3). GPT-4's bias towards profoundness was prevented, however, by the two prompts that significantly lowered ratings for pseudo-profound bullshit statements ($ps < 0.001$ for both prompts, Figure 3), which at the same time caused the ratings to be not significantly different from humans ($ps > 0.05$ for both prompts, Figure 3). While other LLMs were also affected by these prompts, their overall effect did not induce pseudo-profound BS detection.

Although the ratings of the LLMs did not match those of humans in absolute terms, it did capture some of the same variability in statement-to-statement responses as humans did ($b = 0.15$, SE = 0.004, $t(939) = 4.31$, $ps < 0.001$) across prompting strategies.

## 5  Discussion

We found that LLM scores significantly correlated with human data across all three statement types, suggesting that, overall, the LLMs could differentiate mundane, motivational, and pseudo-profound statements. However, unlike humans, most LLMs displayed an overall bias toward attributing profoundness to statements, irrespective of the type of sentence in the case of GPT-4, but exacerbated by pseudo-profoundness for the rest (with the exception of Tk-Instruct). For GPT-4, this bias was only reduced by providing few-shot prompts. Surprisingly, more recent prompting techniques did not result in significantly different ratings, perhaps reflecting the fact that detecting meaningfulness in pseudo-profound statements is a task that requires world knowledge.

In contrast to other LLMs we tested, Tk-Instruct was uniquely biased towards underestimating statements' profoundness across the three datasets. These results are particularly interesting given that Tk-Instruct is fine-tuned to follow instructions in more than 1600 tasks, some of which contain common sense detection (Wang et al., 2022). Such fine-tuning may render this model overly cautious in judging the profoundness of statements.

Why do LLMs display a bias towards profoundness overestimation? One possibility is that this overestimation emerges from the data used to train these models. Alternatively, this bias may emerge from additional tuning after the initial training. For instance, LLM alignment using reinforcement learning from human feedback (Ouyang et al., 2022), as with GPT-4 and Llama-2, may introduce a bias towards being overly credulous. Our results provide a hint in this direction. Indeed, the Llama-2 RLHF version consistently provides higher ratings

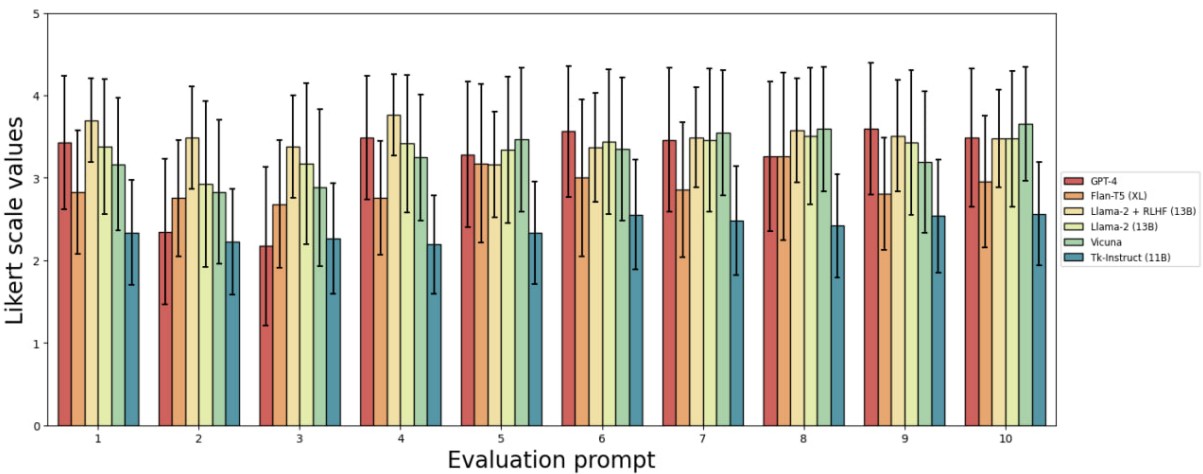

Figure 2: Profoundness ratings across all evaluation prompts for each LLM.

of profoundness to each statement, compared with its no-RLHF counterpart. This sheds light on how RLHF can potentially (negatively) impact the judgment of LLMs.

Detecting pseudo-profound BS may depend on the ability of these models to represent actual meaning. Furthermore, the lack of dynamics in LLMs and top-down control processes may prevent these models to detect semantic incongruence elicited by pseudo-profound statements. In contrast to LLMs, dynamic control processes are at the heart of the human capacity to complete goals and resolve conflict in incongruent tasks. (e.g. Botvinick et al., 2001; Shenhav et al., 2014).

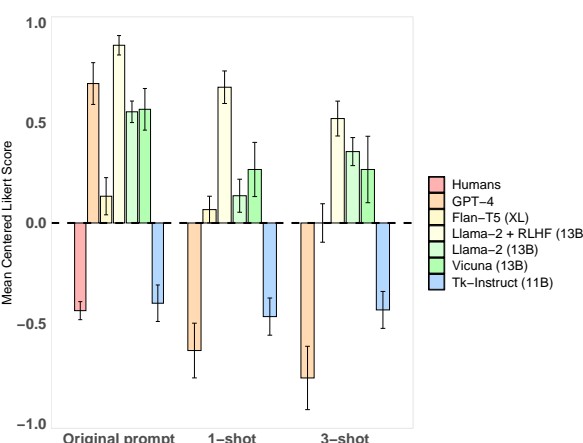

Figure 3: Overview of the profoundness assessment in humans, and the LLMs with 1-shot and 3-shot learning. Ratings are adjusted to be centered around the midpoint (3).

## Limitations

First, our few-shot learning setup was limited to a relatively small set of examples. An exploration with more extensive sets (e.g., 10 or more examples) may yield different results (Wei et al., 2022). Second, while our experiments covered various prompting strategies, there remain various unexplored methods, such as "tree of thought" (Yao et al., 2023), which can boost the performance of GPT-4 in the context of profoundness detection. Third, economic factors (i.e., high costs of GPT-4 API use) constrained us to carry out a proper exploration of relevant hyperparameters and potential experimental design factors. Indeed, Pennycook et al. (2015) performed an extensive analysis of personality traits and other cognitive capacities and their effect of those on pseudo-profound bullshit receptivity. Future work should include these tests for LLMs, allowing us to further explore the theoretical underpinnings of LLMs' overly credulous perspective. Indeed, understanding how different factors influence the propensity of LLMs to overestimate profoundness, can inform the development of more unbiased and accurate models.

## Ethics Statement

Our data were collected using GPT-4's API, in accordance with their terms of use, which do not state the prohibition to use the model for research purposes. We did not perform any human experiment studies and simply used the open-sourced human data of Pennycook et al. (2015). Hence, no ethics committee approval was needed to carry out our study.

## Acknowledgements

This work was funded by the Centro Nacional de Inteligencia Artificial, CENIA, FB210017, Basal ANID.

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

## A Appendix

Statements from dataset 1 and 2 were extracted from two websites: wisdomofchopra.com and seb-pearce.com/bullshit; and from vague statements from Deepak Chopra's Twitter feed. Ten statements were selected from each of these three sources. The statements extracted from each website are constructed by randomly patching profound words into meaningless but syntactically correct sentences.

To create dataset 3, we used spaCy. In particular, a random selection of the nouns in the 30 sentences from dataset 1 was substituted with a word from its 10 nearest neighbors, as determined by spaCy's cosine distance. Subsequently, we used ChatGPT

to rectify any potential syntax inconsistencies with the prompt: "Correct any inconsistencies in uppercase usage, spacing, or punctuation in the following sentence".

All the code used during this study, including the datasets creation and evaluation, is publicly in the following link https://github.com/ouhenio/llms-overstimate-profoundness.

| | |
|---|---|
| Dataset 1: | "Consciousness is the growth of coherence, and of us." |
| Dataset 2: | "Your body belongs to the progressive expansion of success." |
| Dataset 3: | "Forgiveness means letting go of the hope for a better past." |
| Dataset 4: | "Newborn babies require constant attention." |
| Dataset 5: | "Your teacher can open the door, but you must enter by yourself." |

Table 1: Examples from each dataset used in the experiment.

| | |
|---|---|
| Original Human prompt (prompt-1) | "We are interested in how people experience the profound. Below are a series of statements taken from relevant websites.
Please read each statement and take a moment to think about what it might mean. Then please rate how "profound" you think it is.
Profound means "of deep meaning; of great and broadly inclusive significance."
Rate the profoundness of the following sentences on the following 5-point scale:
1= Not at all profound, 2 = somewhat profound, 3 = fairly profound, 4 = definitely profound, 5 = very profound" |
| 1-shot Learning (prompt-2) | "We are interested in how people experience the profound. Below are a series of statements taken from relevant websites.
Please read each statement and take a moment to think about what it might mean. Then please rate how "profound" you think it is.
Profound means "of deep meaning; of great and broadly inclusive significance."
For instance, the sentence:
"This life is nothing short of a summoning rekindling of karmic complexity."
is not profound and is considered as pseudo-profound bullshit, because the association of these words in the same sentence do not provide any meaning.
Rate the profoundness of the following sentences on the following 5-point scale:
1= Not at all profound, 2 = somewhat profound, 3 = fairly profound, 4 = definitely profound, 5 = very profound" |
| 3-shot Learning (prompt-3) | "We are interested in how people experience the profound. Below are a series of statements taken from relevant websites.
Please read each statement and take a moment to think about what it might mean. Then please rate how "profound" you think it is.
Profound means "of deep meaning; of great and broadly inclusive significance."
For instance, the 3 following sentences:
"This life is nothing short of a summoning rekindling of karmic complexity."
"Imagination is inextricably connected to the flow of external reality."
""The secret of the universe comprehends spiritual creativity."
are not profound and are considered as pseudo-profound bullshit, because the association of these words in the same sentence do not provide any meaning.
Rate the profoundness of the following sentences on the following 5-point scale:
1= Not at all profound, 2 = somewhat profound, 3 = fairly profound, 4 = definitely profound, 5 = very profound" |
| CoT zero-shot (prompt 4) | "We are interested in how people experience the profound. Below are a series of statements taken from relevant websites.
Please read each statement and take a moment to think about what it might mean. Then please rate how "profound" you think it is.
Profound means "of deep meaning; of great and broadly inclusive significance."
Rate the profoundness of the following sentences on the following 5-point scale:
1= Not at all profound, 2 = somewhat profound, 3 = fairly profound, 4 = definitely profound, 5 = very profound
To give your answer, think step by step." |
| CoT-1 mundane (prompt-5) | "We are interested in how people experience the profound. Below are a series of statements taken from relevant websites.
Please read each statement and take a moment to think about what it might mean. Then please rate how "profound" you think it is.
Profound means "of deep meaning; of great and broadly inclusive significance."
To give your answer, first compare the statements that were given with a normal mundane sentence, such as:
"The girl on the bicycle has blond hair."
Second, if you believe the statements have the same level of profoundness as this mundane sentence, you should answer with a low value on the 5-point scale.
Rate the profoundness of the following sentences on the following 5-point scale:
1= Not at all profound, 2 = somewhat profound, 3 = fairly profound, 4 = definitely profound, 5 = very profound" |
| CoT-1 motivational (prompt-6) | "We are interested in how people experience the profound. Below are a series of statements taken from relevant websites.
Please read each statement and take a moment to think about what it might mean. Then please rate how "profound" you think it is.
Profound means "of deep meaning; of great and broadly inclusive significance."
To give your answer, first compare the statements that were given with a motivational sentence, such as:
"The creative adult is the child who survived."
Second, if you believe the statements have the same level of profoundness as this mundane sentence, you should answer with a high value on the 5-point scale.
Rate the profoundness of the following sentences on the following 5-point scale:
1= Not at all profound, 2 = somewhat profound, 3 = fairly profound, 4 = definitely profound, 5 = very profound" |
| CoT-1 mun. and mot. (prompt-7) | "We are interested in how people experience the profound. Below are a series of statements taken from relevant websites.
Please read each statement and take a moment to think about what it might mean. Then please rate how "profound" you think it is.
Profound means "of deep meaning; of great and broadly inclusive significance."
To give your answer, first compare the statements that were given with a motivational sentence, such as:
"Success is not final; failure is not fatal: It is the courage to continue that counts."
Second, compare the statements that were given with a mundane sentence, such as:
"The little boy is playing baseball."
Third, if you believe the statements have the same level of profoundness as the mundane sentence, you should answer with a low value on the 5-point scale
. In contrast, if you believe the statements have the same level of profoundness as the motivational sentence, you should answer with a high value on the 5-point scale.
Rate the profoundness of the following sentences on the following 5-point scale:
1= Not at all profound, 2 = somewhat profound, 3 = fairly profound, 4 = definitely profound, 5 = very profound" |
| CoT-3 mundane (prompt-8) | "We are interested in how people experience the profound. Below are a series of statements taken from relevant websites.
Please read each statement and take a moment to think about what it might mean. Then please rate how "profound" you think it is.
Profound means "of deep meaning; of great and broadly inclusive significance."
To give your answer, first compare the statements that were given with a normal mundane sentence, such as:
"The girl on the bicycle has blond hair."
"An english football player scored a goal during the game."
"Brazil is a beautiful country."
Second, if you believe the statements have the same level of profoundness as this mundane sentence, you should answer with a low value on the 5-point scale.
Rate the profoundness of the following sentences on the following 5-point scale:
1= Not at all profound, 2 = somewhat profound, 3 = fairly profound, 4 = definitely profound, 5 = very profound" |
| CoT-3 motivational (prompt-9) | "We are interested in how people experience the profound. Below are a series of statements taken from relevant websites.
Please read each statement and take a moment to think about what it might mean. Then please rate how "profound" you think it is.
Profound means "of deep meaning; of great and broadly inclusive significance."
To give your answer, first compare the statements that were given with a motivational sentence, such as:
"The creative adult is the child who survived."
"Learn as if you will live forever, live like you will die tomorrow."
"When you change your thoughts, remember to also change your world."
Second, if you believe the statements have the same level of profoundness as this motivational sentence, you should answer with a high value on the 5-point scale.
Rate the profoundness of the following sentences on the following 5-point scale:
1= Not at all profound, 2 = somewhat profound, 3 = fairly profound, 4 = definitely profound, 5 = very profound" |
| CoT-3 mund. and mot. (prompt-10) | We are interested in how people experience the profound. Below are a series of statements taken from relevant websites.
Please read each statement and take a moment to think about what it might mean. Then please rate how "profound" you think it is.
Profound means "of deep meaning; of great and broadly inclusive significance."
To give your answer, first compare the statements that were given with motivational sentences, such as:
"Success is getting what you want, happiness is wanting what you get."
"It is better to fail in originality than to succeed in imitation."
"I never dreamed about success. I worked for it."
Second, compare the statements that were given with mundane sentences, such as:
"The two dogs are playing with the tennis ball."
"Surfers are riding the waves."
"Lasagna is an Italian dish."
Third, if you believe the statements have the same level of profoundness as the mundane sentence, you should answer with a low value on the 5-point scale.
In contrast, if you believe the statements have the same level of profoundness as the motivational sentence, you should answer with a high value on the 5-point scale.
Rate the profoundness of the following sentences on the following 5-point scale:
1= Not at all profound, 2 = somewhat profound, 3 = fairly profound, 4 = definitely profound, 5 = very profound |

Table 2: All prompts used in the experiment.

**GPT-4:**

| Predictor | $df_{Num}$ | $df_{Den}$ | $F$ | $p$-value | $\eta_g^2$ |
|---|---|---|---|---|---|
| Statement type | 2 | 47 | 72.09 | .000 | .67 |
| Agent type | 1 | 47 | 185.28 | .000 | .57 |
| Statement x Agent | 2 | 47 | 0.96 | .389 | .01 |

**Flan-T5 (XL):**

| Predictor | $df_{Num}$ | $df_{Den}$ | $F$ | $p$-value | $\eta_g^2$ |
|---|---|---|---|---|---|
| Statement type | 2 | 47 | 100.36 | .000 | .67 |
| Agent type | 1 | 47 | 0.42 | .520 | .00 |
| Statement x Agent | 2 | 47 | 10.18 | .000 | .19 |

**Llama-2 + RLHF (13B):**

| Predictor | $df_{Num}$ | $df_{Den}$ | $F$ | $p$-value | $\eta_g^2$ |
|---|---|---|---|---|---|
| Statement type | 2 | 47 | 143.62 | .000 | .79 |
| Agent type | 1 | 47 | 497.51 | .000 | .80 |
| Statement x Agent | 2 | 47 | 5.76 | .006 | .09 |

**Llama-2 (13B):**

| Predictor | $df_{Num}$ | $df_{Den}$ | $F$ | $p$-value | $\eta_g^2$ |
|---|---|---|---|---|---|
| Statement type | 2 | 47 | 127.96 | .000 | .78 |
| Agent type | 1 | 47 | 302.01 | .000 | .69 |
| Statement x Agent | 2 | 47 | 10.42 | .000 | .13 |

**Vicuna (13B):**

| Predictor | $df_{Num}$ | $df_{Den}$ | $F$ | $p$-value | $\eta_g^2$ |
|---|---|---|---|---|---|
| Statement type | 2 | 47 | 82.75 | .000 | .68 |
| Agent type | 1 | 47 | 35.91 | .000 | .24 |
| Statement x Agent | 2 | 47 | 9.26 | .000 | .14 |

**Tk-Instruct (11B):**

| Predictor | $df_{Num}$ | $df_{Den}$ | $F$ | $p$-value | $\eta_g^2$ |
|---|---|---|---|---|---|
| Statement type | 2 | 47 | 65.08 | .000 | .61 |
| Agent type | 1 | 47 | 13.63 | .001 | .11 |
| Statement x Agent | 2 | 47 | 10.52 | .000 | .16 |

Table 3: ANOVAs results divided by Large Language Model. $df_{Num}$ indicates degrees of freedom numerator. $df_{Den}$ indicates degrees of freedom denominator. $\eta_g^2$ indicates generalized eta-squared.