# OpenReview forum: "Large Language Models are biased to overestimate profoundness"
_EMNLP/2023/Conference — EMNLP 2023 Main_

### Official Review · Reviewer_JFhv · 2023-07-26

**Typos Grammar Style And Presentation Improvements:** 1. Figure 2 caption should be standalone
**Soundness:** 3

**Excitement:**

3: Ambivalent: It has merits (e.g., it reports state-of-the-art results, the idea is nice), but there are key weaknesses (e.g., it describes incremental work), and it can significantly benefit from another round of revision. However, I won't object to accepting it if my co-reviewers champion it.

**Missing References:**

Bullshit detection falls into the realm of pragmatics and, in particular, pragmatic deception.

Missing relevant work on bullshit and LMs, which proposes an alternative pipeline for bs detection (Deck, 2023 https://journals.uic.edu/ojs/index.php/dad/article/view/12690).

Missing other explicit references to LMs and pragmatics, crucially Hu et al. 2023 (https://arxiv.org/abs/2212.06801) which measures human-LM alignment on pragmatic reasoning tasks, as well as Ruis et al. 2022 (https://arxiv.org/abs/2210.14986), etc.

**Paper Topic And Main Contributions:**

This paper is one in a line of contributions evaluating LMs against human judgments, in this case, applied to pseudo-profound bullshit.

Pseudo-profound bullshit is a type of language that appears meaningful, but that lacks substance. The authors examine the extent to which GPT-4 matches human judgments on pseudo-profound bs, finding that (1) although statement profoundness judgments are correlated between GPT-4 and humans, (2) GPT-4 is biased towards profoundness. They then evaluate the effect of prompting method, showing that (3) chain-of-thought prompting methods don't significantly affect profoundness rankings; (4) few-shot learning allows GPT-4 to rate statements more similar to humans.

**Questions For The Authors:**

A. l170: How is an answer determined? E.g., does GPT-4 have to exactly match a number on the Likert scale, or is it by highest probability assigned?

B. l280: What is b?

**Reasons To Accept:**

The strengths of the paper are that it, to my knowledge,

1. Answers a novel research question: while not the first to tie pseudo-profound bullshit to NLP (Deck, 2023), this paper is the first to present results on LLMs' judgments of profoundness. In the current media landscape where LLMs are being deployed, it is likely that they encounter pseudo-profound bullshit, and it is thus important to understand their behavior.

2. GPT-4 prompts are reproducible: the authors provide all prompts used in experiments.

**Reasons To Reject:**

My main reservations accepting the paper are regarding scope and discussion:

__Scope__ As mentioned in the Limitations section, while the authors make a claim about LLMs broadly in the title, they only test on one model, GPT-4. In order for the claims to have maximal impact, results need to be demonstrated on a wider range of model families and sizes. This would resolve, e.g., some of the hypotheses posed at the end of the paper about the impact of RLHF on profoundness ratings. Moreover, given the small dataset sizes and free compute available on, e.g., Colab, it seems feasible to reproduce experiments on a wider set of models such as OPT, cf. Hu, et al. 2023 for similar experiments on multiple choice questions, performed on several different models.

__Discussion__ While the paper is quite results-dense, it lacks a broader discussion that relates human and LM pragmatics/social cognition. The paper heavily cites Pennycook et al. 2015, which proposes two cognitive mechanisms underlying bullshit detection in humans-- a first, general bias towards overestimating profoundness, then a failure in downstream "conflict monitoring". Along these lines, this paper can be made more impactful by (1) making an explicit connection to the cognitive theories proposed in the former; (2) making a statement about practical implications of results-- why does it matter that GPT-4 overestimates profoundness?

While the research question is promising, results are only presented on one LM and there lacks a broader discussion and tie-in to related work in pragmatics. This limits the impact of the paper.

Minor:
While the authors focus on GPT-4, they mention BERT in Related Work. I would recommend scoping this work to causal language models only (or GPT family only) and omitting related work on BERT altogether, as MLM and CLM are inherently different objectives.

**Reproducibility:**

5: Could easily reproduce the results.

**Reviewer Confidence:**

4: Quite sure. I tried to check the important points carefully. It's unlikely, though conceivable, that I missed something that should affect my ratings.

---

> ### Author Rebuttal · Authors · 2023-08-29
>
> We thank the reviewer for her/his valuable comments, and we hope that the following replies convince her/him of the merits of our work.
>
> As for reviewer T4Lt, the first issue raised is that of the scope. Our goal was not to systematically compare distinct models, but principally to provide a method to test the limits of GPT-4. Although our work is based on a series of studies evaluating the similarity between human and LLMs judgments, we believe that our work is unique in that the statements used to assess GPT-4 belong to a very specific type, i.e., pseudo-profound bullshit. Our work thus helps delineate the limits of GPT-4. However, we agree with the reviewer that our work cannot be generalized across LLMs. To further justify our claims that such an effect is pervasive to LLMs in general, we ran the same experiment using several other models: Flan-T5 XL, Llama-2 (13B), Llama-2 + RLHF (13B), Vicuna (13B), and Tk-Instruct (11B). Our results reveal an interesting pattern. All models overestimate the profoundness of pseudo-profound bullshit statements, except Tk-instruct. Specifically, Flan-T5 XL, Llama-2 (13B), Llama-2 + RLHF (13B), and Vicuna (13B) display the same qualitative pattern as that of GPT-4: they overestimate the profoundness of bullshit statements, both when compared with humans and with the midpoint level of the scale. In contrast, Tk-Instruct, rate pseudo-profound bullshit statements similar to humans. Yet, Tk-Instruct also displays a strong bias to underestimate the profoundness of motivational statements compared with humans. Hence, this model is generally biased to underestimate the profoundness of statements. These results are interesting given that Tk-instruct is fine-tuned on following instructions in more than 1600 tasks, some of which contain common sense detection. Such fine-tuning may render this model overly cautious in terms of profoundness rating.
>
> Furthermore, on top of broadening the scope of the paper by testing more models, adding those models also allows us to test the effect of RLHF on the statement ratings directly. Indeed, this can be done globally by contrasting GPT-4/Llama-2 + RLHF vs. all other models; but also locally, by directly comparing Llama-2 with Llama-2 + RLHF. Note that the only difference between those models is the addition of RLHF. At the global level, our results show that models with RLHF tend to further overestimate the profoundness of BS and motivational statements compared with models with no RLHF. Crucially, at the local level, we observe the same trend: when compared with its counterparts with no RLFH, Llama-2+RLHF overestimates the profoundness of BS (and motivational) statements.
>
> Such results imply that, although LLMs have a general propensity to overestimate profoundness of BS statements, this bias is enhanced by the inclusion of RLFH. Therefore, these results may emerge from the additive effects of (i) the inability of LLMs to dissociate deceptiveness from syntactic structure of sentences; which could explain why n-shot learning generates better performance, as LLMs may simply recognize a type of “awe-inspiring” words and make judgments based on this semantic similarity, and (ii) that RLHF may have introduced a bias towards being overly credulous. Moreover, instruction fine-tuning may have a distinct effect by making these models overly cautious when judging the profoundness or meaningfulness of statements.
>
> Regarding the discussion, we did not explicitly relate our results to cognitive theories, nor did we discuss the implications of these results for GPT-4. We purposely avoided discussing these issues as we wanted the scope of this paper to be focused on a novel and original way to test the limits of LLMs, outside the realm of typical cognitive abilities tests or sensical sentence-based judgments (e.g., Hu et al. (2023)). We still believe this is what our paper should focus on, and that such a scope is of great interest for the field. Yet, we agree with the reviewer that discussing the link with previous cognitive theories and the implications of this inability would strengthen the paper.
>
> With respect to cognitive theories: It may be that the lack of dynamics in LLMs and top-down control processes, does not allow these models to detect semantic incongruence. Such control processes are at the heart of many cognitive control models in cognitive science, and allow to resolve conflict in such tasks (e.g., Botvinick et al., 2004; Shenhav et al., 2014). Moreover, Pennycook et al. (2015) performed extensive analysis of personality traits and other cognitive capacities, and the effect of those on pseudo-profound bullshit receptivity. Future work should  include these tests for LLMs, allowing to further explore the theoretical underpinnings of LLMs overly credulous perspective. Our CR version will relate conflict monitoring perspectives with that of Pennycook et al. (2015), discuss future work, and therefore further broaden our discussion.
>
> With respect to implications: First, our result should come as a warning in case these models are used within contexts that make use of awe-inspiring words (e.g., quantum physics). Indeed, these models may spew out nonsensical sentences, while being conformed of their meaningfulness. Second, although humans do a better job at detecting deceptive language, specific biases can be injected within LLMs through the use of RLHF, making these models overly credulous. Finally, whereas the ability to perform specific cognitive abilities can increase through scaling, the inability to detect deceptive language may emerge from core features of LLMs, such as their next-word-prediction objective or their lack of recurrent dynamics. Indeed, even GPT-4 (with presumably more than a trillion parameters) overestimates the profoundness of BS statements.
>
> Our CR version will address the two main comments raised by the reviewer. The graphs will include the results of all LLMs, thereby expanding the scope of our paper to LLMs. The reviewer can access the visualization of our revised 3 figures on the anonymized GitHub repository at this address: https://github.com/llms-are-biased/to-overestimate-profoundess (this anonymized GitHub also contains the code to extract the data from the LLMs; increasing the reproducibility of our work). Moreover, for the purpose of data visualization (given the significant amount of LLMs that we have added to this work) all figures now represent the data as bar plots (cfr. Hu et al., 2023). The appendices will contain the same statistics we performed on GPT-4, but for all the other LLMs we have tested. These statistics will be shown in the form of a table (accessible on the previously mentioned GitHub repository address), which essentially shows that the observed results for GPT-4 hold across the board of LLMs (Except Tk-Instruct). Furthermore, we will explicitly address why the inability to detect deceptive language is an issue in LLMs (as stated in the previous paragraph), and explicitly link our results to the cognitive theories proposed by Pennycook et al. (2015; also as explained above). Finally, as suggested by the reviewer, we will focus this work on causal language models only and omit related work on BERT, incorporate all the missing references (Hu et al., 2023; Ruis et al., 2022; Deck, 2023), correct the typos, and incorporate the suggested presentation improvements.
>
> Questions For The Authors:
> A. l170: How is an answer determined? E.g., does GPT-4 have to exactly match a number on the Likert scale, or is it by highest probability assigned?
>
> We set the number of output tokens to 1.
>
> B. l280: What is b?
>
> b stands for the regression coefficient of the linear regression predicting human ratings from GPT-4 ratings.
>
> We hope our replies address the comments raised by the reviewer, and that the substantial amount of work we have put into this revision can convince them of the added value of this research for the NLP field.
>
> References:
> -Botvinick, Matthew M., et al. "Conflict monitoring and cognitive control." Psychological review 108.3 (2001): 624.
> -Shenhav, Amitai, et al. "Anterior cingulate engagement in a foraging context reflects choice difficulty, not foraging value." Nature neuroscience 17.9 (2014): 1249-1254.
> -Hu, Jennifer, et al. "A fine-grained comparison of pragmatic language understanding in humans and language models." arXiv preprint arXiv:2212.06801 (2022).

---

### Official Review · Reviewer_W7L6 · 2023-08-03

**Soundness:** 3

**Excitement:**

3: Ambivalent: It has merits (e.g., it reports state-of-the-art results, the idea is nice), but there are key weaknesses (e.g., it describes incremental work), and it can significantly benefit from another round of revision. However, I won't object to accepting it if my co-reviewers champion it.

**Paper Topic And Main Contributions:**

In this paper, they evaluate GPT-4’s ability to judge the profoundness of mundane, motivational, and pseudo-profound statements. They found a significant statement-to-statement correlation between GPT-4 and humans, irrespective of the type of statements and the prompting technique used. GPT-4 systematically overestimates the profoundness of non-sensical statements in contrast with humans.

**Reasons To Accept:**

This paper starts from an interesting perspective to probe the ability of GPT4 and shows some observations by prompting GPT4. I suggest it can be accepted into Findings.

**Reasons To Reject:**

Basically this paper follows the previous work heavily. So I think the novelty is not enough. Based on some interesting results from the experiments, it may be accepted to findings.

**Reproducibility:**

2: Would be hard pressed to reproduce the results. The contribution depends on data that are simply not available outside the author's institution or consortium; not enough details are provided.

**Reviewer Confidence:**

2: Willing to defend my evaluation, but it is fairly likely that I missed some details, didn't understand some central points, or can't be sure about the novelty of the work.

---

> ### Author Rebuttal · Authors · 2023-08-29
>
> We thank the reviewer for her/his appraisal of our work, and relevant comments. We address the novelty of our work in what follows.
>
> Previous studies focused either on circumscribing the limits of typical cognitive abilities (e.g., reasoning) in LLMs, or their ability to reply to distinct types of judgments as humans (Hu et al., 2023). The crucial distinction in our work lies in that we specifically focus on the ability of GPT-4 to detect deceptive language. Therefore, our work targets the communication abilities of these models. In particular, compared with previous work (see Hu et al., 2023), we specifically focus on deceptive language that aims to create the false impression of depth, instantiated as syntactically correct yet pseudo-profound bullshit sentences. To the best of our knowledge, no other work has probed GPT-4 with this type of deceptive language in a systematic way, while exploring distinct state-of-the-art prompting techniques. Thus, in light of our results, we do believe that our work makes for an important and original contribution to the field.
>
> Moreover, based the reviewers’ feedback, we have now broadened the scope of our paper and ran the same experiment using several other models: Flan-T5 XL, Llama-2 (13B), Llama-2 + RLHF (13B), Vicuna (13B), and Tk-Instruct (11B). Our results reveal an interesting pattern. All models overestimate the profoundness of pseudo-profound bullshit statements, except Tk-instruct. Specifically, Flan-T5 XL, Llama-2 (13B), Llama-2 + RLHF (13B), and Vicuna (13B) display the same qualitative pattern as that of GPT-4: they overestimate the profoundness of bullshit statements, both when compared with humans and with the midpoint level of the scale. In contrast, Tk-Instruct, rates pseudo-profound bullshit statements similar to humans. Yet, Tk-Instruct also displays a strong bias to underestimate the profoundness of motivational statements compared with humans. Hence, this model is generally biased to underestimate the profoundness of statements. These results are interesting given that Tk-instruct is fine-tuned to follow instructions in more than 1600 tasks, some of which contain common sense detection. Such fine-tuning may render this model overly cautious in terms of profoundness rating.
>
> Furthermore, on top of broadening the scope of the paper by testing more models, we can directly test the effect of RLHF on the statement ratings. Indeed, this can be done globally by contrasting GPT-4 and Llama-2 + RLHF vs. all other models; but also locally, by directly comparing Llama-2 with Llama-2 + RLHF. Note that the only difference between those models is the addition of RLHF. At the global level, our results show that models with RLHF tend to further overestimate the profoundness of BS and motivational statements compared with models with no RLHF. Crucially, at the local level, we observe the same trend: when compared its counterpart with no RLFH, Llama-2+RLHF overestimates the profoundness of BS (and motivational) statements. Such results imply that, although LLMs have a general propensity to overestimate the profoundness of BS statements, this bias is enhanced by the inclusion of RLFH. Therefore, these results may emerge from the additive effects of (i) the inability of LLMs to dissociate deceptiveness from syntactic structure of sentences; which could explain why n-shot learning generates better performance, as LLMs may simply recognize a type of “awe-inspiring” words and make judgments based on this semantic similarity, and (ii) that RLHF may have introduced a bias towards being overly credulous. Moreover, instruction fine-tuning may have a distinct effect by making these models overly cautious when judging the profoundness or meaningfulness of statements.
>
> Our CR version will address the novelty comment raised by the reviewer. Moreover, our graphs will include the results of all LLMs, thereby expanding the scope of our paper. The reviewer can access the visualization of our revised 3 figures on the anonymized github repository at this address: https://github.com/llms-are-biased/to-overestimate-profoundess (this anonymized GitHub also contains the code to extract the data from the LLMs; increasing the reproducibility of our work). This repository contains all the prompts (also available in the appendix) and the datasets (also available in the supporting information of Pennycook et al. (2015). For the purpose of data visualization (given the significant amount of LLMs that we have added to this work) all figures now represent the data as bar plots (cfr. Hu et al., 2023). The appendices will contain the same statistics we performed on GPT-4, but for all the other LLMs we have tested. These statistics will be shown in the form of a table (accessible on the previously mentioned github repository address), which essentially shows that the observed results for GPT-4 hold across the board of LLMs (except for Tk-Instruct).
>
> We hope these changes can warrant the publication of our work, which we believe tackles a timely issue and can be of great interest to researchers in NLP.
>
> References: Hu, Jennifer, et al. "A fine-grained comparison of pragmatic language understanding in humans and language models." arXiv preprint arXiv:2212.06801 (2022).

---

### Official Review · Reviewer_T4Lt · 2023-08-04

**Soundness:** 4

**Excitement:**

3: Ambivalent: It has merits (e.g., it reports state-of-the-art results, the idea is nice), but there are key weaknesses (e.g., it describes incremental work), and it can significantly benefit from another round of revision. However, I won't object to accepting it if my co-reviewers champion it.

**Paper Topic And Main Contributions:**

This paper investigates how GPT-4 rates statements for profoundness. The core empirical observation is that humans can be susceptible to what is termed “pseudo-profound bullshit” – statements that may seem profound at first, but sustained reflection can expose as meaningless (e.g., “your body belongs to the progressive expansion of success”). The main contribution of this work is a demonstration that GPT-4 is more prone to viewing such statements as profound as compared to humans. An additional contribution is the datasets used to evaluate the modes.

**Questions For The Authors:**

A. What is the meaning of N=20 for GPT-4 on line 142?

**Reasons To Accept:**

The paper is well written and the problem interesting. The focus on language that is not “aimed at efficiently communicating information” is timely and important. Further, the experiments are reasonably constructed and analysis well done.

**Reasons To Reject:**

The main weaknesses of the paper are: i) limited spread of models investigated, and ii) limited framing for why the problem as investigated is a problem for GPT-4. For i), the paper only investigates GPT-4, and so limits the scope of the claims to a model with very little transparency. As such, it is difficult to extrapolate from this paper how other models may behave and the conditions under which these effects obtain. The discussion mentions that RLHF “might have introduced a bias towards being overly credulous”. This claim could be addressed by exploring other models, thus making the results more impactful.

For ii), I take the work to be suggesting that for humans, pseudo-profound bullshit is a particular type of statement that utilizes limitations in our reasoning to trick people. The investigation of GPT-4 demonstrates that may be more susceptible to such illusions. However, the work lacks discussion of why this poses a particular problem for these models. I can imagine that a model with a propensity to generate pseudo-profound bullshit would cause problems, as it would mislead people. Articulating why we should be concerned by models being misled by such statements would strengthen the paper.

**Reproducibility:**

4: Could mostly reproduce the results, but there may be some variation because of sample variance or minor variations in their interpretation of the protocol or method.

**Reviewer Confidence:**

4: Quite sure. I tried to check the important points carefully. It's unlikely, though conceivable, that I missed something that should affect my ratings.

**Typos Grammar Style And Presentation Improvements:**

It might be nice to include an example of pseudo-profound bullshit in the introduction (perhaps moving Table 1 to the main text once you have more space).

---

> ### Author Rebuttal · Authors · 2023-08-29
>
> We thank the reviewer for their valuable comments. In this paper, we specifically focused on GPT-4 given the claims made regarding its ability to demonstrate sparks of AGI. But also, and perhaps even more crucial, due to the massive industrial applications we are witnessing based on this model. For those reasons, we believe it is important to test the limitation of GPT-4 (taken as the sota LLM model). And, in our opinion, any such limitation must be considered an important addition to the literature.
>
> The evaluation of LLMs typically focuses on tests that target traditional cognitive abilities (e.g., reasoning, planning, spatial navigation…). Our aim in this paper was to take a different route. Here, we particularly focused on the ability of LLMs to detect deceptive language, and therefore target the communication abilities of these models. We believe that our results showing the lack of ability for GPT-4 to detect deceptive language is, standing on its own, of great value to the NLP field, not only at the scientific level, but also at the practical one.
>
> Nonetheless, we agree with both comments raised by the reviewer:
>
>
> **Regarding the scope**
>
> First, the scope of our paper was limited to GPT-4, but one could make the claim that this extends to the open AI GPT model family (as GPT-4 is the SOTA model). To further justify our claims that such an effect is pervasive to LLMs in general, we ran the same experiment using several other models: Flan-T5 XL, Llama-2 (13B), Llama-2 + RLHF (13B), Vicuna (13B), and Tk-Instruct (11B). Our results reveal an interesting pattern. All models overestimate the profoundness of pseudo-profound bullshit statements, except Tk-instruct. Specifically, Flan-T5 XL, Llama-2 (13B), Llama-2 + RLHF (13B), and Vicuna (13B) display the same qualitative pattern as that of GPT-4: they overestimate the profoundness of bullshit statements, both when compared with humans and with the midpoint level of the scale. In contrast, Tk-Instruct, rates pseudo-profound bullshit statements similar to humans. Yet, Tk-Instruct also displays a strong bias to underestimate the profoundness of motivational statements compared with humans. Hence, this model is generally biased to underestimate the profoundness of statements. These results are interesting given that Tk-instruct is fine-tuned to follow instructions in more than 1600 tasks, some of which contain common sense detection. Such fine-tuning may render this model overly cautious in terms of profoundness rating.
>
> Furthermore, on top of broadening the scope of the paper by testing more models, adding those models also allows us to test the effect of RLHF on the statement ratings directly. Indeed, this can be done globally by contrasting GPT-4/Llama-2 + RLHF vs. all other models, but also locally by directly comparing Llama-2 with Llama-2 + RLHF. Note that the only difference between those models is the addition of RLHF. At the global level, our results show that models with RLHF tend to overestimate the profoundness of BS and motivational statements even more compared with models with no RLHF. Crucially, at the local level, we observe the same trend: when compared with its counterparts with no RLFH, Llama-2+RLHF overestimates the profoundness of BS (and motivational) statements.
>
> Such results imply that, although LLMs have a general propensity to overestimate the profoundness of BS statements, this bias seems to be enhanced by the inclusion of RLFH. Therefore, these results may emerge from the additive effects of (i) the inability of LLMs to dissociate deceptiveness from non-deceptive sentences, which could explain why n-shot learning generates better performance, as LLMs may simply recognize a type of “awe-inspiring” words and make judgments based on this semantic similarity, and (ii) that RLHF may have introduced a bias towards being overly credulous. Moreover, instruction fine-tuning may have a distinct effect by making these models overly cautious when judging the profoundness or meaningfulness of statements.
>
> **Regarding limited framing**
>
> Second, we did not explicitly state why the inability to detect deceptive language would be a problem for GPT-4. Initially, we purposely avoided discussing this issue as we wanted the scope of this paper to be focused on a novel and original way to test the limits of LLMs, outside the realm of typical cognitive abilities tests, and focus on using syntactically correct but pseudo-profound sentences. We still believe this is what our paper should focus on, and that such a scope is of great interest for the field. Yet, we agree with the reviewer that discussing the implications of this inability would strengthen the paper. Several potential limitations of this inability come to mind. First, our result should come as a warning in case these models are used within contexts that make use of awe-inspiring words (e.g., quantum physics). Indeed, these models may spew out nonsensical sentences while being conformed with regards to their meaningfulness. Hence, this overestimation of profoundness might be an issue in any application scenario. Second, although humans do a better job at detecting deceptive language, specific biases can be injected within LLMs through the use of RLHF, making these models overly credulous, and thus more easily compliant. Finally, whereas the ability to perform specific cognitive abilities can increase through scaling, the inability to detect deceptive language may emerge from core features of LLMs, such as their next-word-prediction objective or their lack of recurrent dynamics. Indeed, even GPT-4 (with presumably more than a trillion parameters) overestimates the profoundness of BS statements.
>
> Our CR version will address these two main comments raised by the reviewer. The graphs will include the results of all LLMs, thereby expanding the scope of our paper to LLMs. The reviewer can access the visualization of our revised 3 figures on the anonymized GitHub repository at this address: https://github.com/llms-are-biased/to-overestimate-profoundess (this anonymized GitHub also contains the code to extract the data from the LLMs; increasing the reproducibility of our work). Moreover, for the purpose of data visualization (given the significant amount of LLMs that we have added to this work), all figures are now represented as bar plots (cfr. Hu et al., 2023; this allows us to increase the readability of the figures compared with violin plots). The appendices will contain the same statistics we performed on GPT-4, but for all the other LLMs we have tested. These statistics will be shown in the form of a table (accessible on the previously mentioned GitHub repository address), which essentially shows that the observed results for GPT-4 hold across the board of LLMs (except for Tk-instruct). Furthermore, we will explicitly address why the inability to detect deceptive language is an issue in LLMs (as stated in the previous paragraph), and we will include an example of pseudo-profound bullshit in the introduction.
>
> Lastly, N=20 is a typical notation used in the field of psycholinguistics and refers to the amount of “participants” in the study. In our case, to perform statistical analyses, we ran GPT-4 20 times; hence, N=20. The same number was also used for all the other LLMs. We will clarify this in the CR submission.
>
> We hope our replies and these changes can convince the reviewer of the novelty, extended scope, and implications of our work, and hope that addressing their comments can warrant the acceptance of our work.
>
> References: Hu, Jennifer, et al. "A fine-grained comparison of pragmatic language understanding in humans and language models." arXiv preprint arXiv:2212.06801 (2022).

---

### Meta-Review · Area_Chair_ooV9 · 2023-09-12

**Recommendation:** 4

**Metareview:**

The authors evaluate GPT-4 (and on request of two reviewers also other models) on the task of judging the "profoundness" of mundane or pseudo-profound statements. While they find a correlation between human scores and model scores, the models tend to give higher profoundness ratings to  statements than humans. The paper compares different prompting techniques, which all exhibit the reported problem.
Two of the reviewers appreciate the importance of the issue that is being addressed; one of the reviewers is worried about a lack of novelty, but this worry is not substantiated by the reviewer -- instead, the task of judging profoundness / detecting lack of substance is a rather underresearched problem.

---

### Decision · Program_Chairs · 2023-10-07

**Decision:**

Accept-Main

**Comment:**

The authors evaluate GPT-4 (and on request of two reviewers also other models) on the task of judging the "profoundness" of mundane or pseudo-profound statements. While they find a correlation between human scores and model scores, the models tend to give higher profoundness ratings to  statements than humans. The paper compares different prompting techniques, which all exhibit the reported problem.
Two of the reviewers appreciate the importance of the issue that is being addressed; one of the reviewers is worried about a lack of novelty, but this worry is not substantiated by the reviewer -- instead, the task of judging profoundness / detecting lack of substance is a rather underresearched problem.